# Large impact cratering during lunar magma ocean solidification

K. Miljković 📷 [1✉], M. A. Wieczorek[2], M. Laneuville 📷 [3], A. Nemchin[1], P. A. Bland[1] & M. T. Zuber 📷 [4]

The lunar cratering record is used to constrain the bombardment history of both the Earth and the Moon. However, it is suggested from different perspectives, including impact crater dating, asteroid dynamics, lunar samples, impact basin-forming simulations, and lunar evolution modelling, that the Moon could be missing evidence of its earliest cratering record. Here we report that impact basins formed during the lunar magma ocean solidification should have produced different crater morphologies in comparison to later epochs. A low viscosity layer, mimicking a melt layer, between the crust and mantle could cause the entire impact basin size range to be susceptible to immediate and extreme crustal relaxation forming almost unidentifiable topographic and crustal thickness signatures. Lunar basins formed while the lunar magma ocean was still solidifying may escape detection, which is agreeing with studies that suggest a higher impact flux than previously thought in the earliest epoch of Earth-Moon evolution.

[1] Curtin University, School of Earth and Planetary Science, Space Science and Technology Centre, Perth, WA, Australia. [2] Université Côte d'Azur, Observatoire de la Côte d'Azur, CNRS, Laboratoire Lagrange, Nice, France. [3] Earth-Life Science Institute, Tokyo, Japan. [4] Massachusetts Institute of Technology, Cambridge, MA, USA. ✉email: katarina.miljkovic@curtin.edu.au

Understanding impact bombardment and the cratering record from the earliest epochs of solar system history is imperative for completing the story of how planets formed and evolved. The long-standing Moon-formation theory whereby a giant impact occurred with proto-Earth implies that the young Moon formed with a global magma ocean[1–7]. The timeframe for the solidification of the lunar magma ocean (LMO) varies significantly between calculations[8,9], from within a few Myr[10] to up to ~200 Myr[1–3,11,12], but could also have varied regionally for up to ~500 Myr[13,14]. Radiogenic lunar crustal ages span from 4.47 Ga to 4.31 Ga, which falls broadly within this range, and the age of the giant impact has been estimated to have occurred at ~4.54–4.425 Ga[1,9,10,15]. Microstructural analyses of mineral assemblages extracted from Apollo samples, such as zircons with shock deformation features associated with an impact event[16], have suggested that a number of large impacts could have occurred during the first ~200 Myr of lunar history, however, no clear identification of the source impact basins has yet been made. Comparisons of the expected impact flux with the current cratering record in the oldest, pre-Nectarian, epoch on the Moon have suggested that the cratering record from this period is incomplete[17,18]. A recent reconstruction of the late-accretion history of the Moon based on impact-delivered siderophile elements has suggested that there could have been as many as 200 basin-forming impacts that formed before 4.35 Ga that are unaccounted for in the current lunar cratering record[19].

The topographic signatures of the oldest basins could have been degraded by subsequent impact bombardment[20,21]. Gravity data acquired by the Gravity Recovery and Interior Laboratory (GRAIL) mission[22], however, have shown that the large and stratigraphically oldest pre-Nectarian impact basins also have muted subsurface crustal signatures compared to the younger Nectarian and Imbrian impact basins[23,24]. Furthermore, while the Nectaris and Orientale basins have visible multi-ring topographic features, the pre-Nectarian impact basins may have only one suspected ring/rim (as indicated by arrows in Fig. 1a)[24]. Figure 1a shows three of the oldest pre-Nectarian basins[25,26] that exhibit

less prominent crustal thinning compared to younger basins of likely similar size, such as the Nectaris and Orientale basins (Fig. 1b). Though the sizes of these basins are similar, based on the diameter of their previously mapped main topographic rings, the older pre-Nectarian basins have a relatively thicker crust in the centre of the basin and also lack the distinct crustal thickening between 1 and 2 main rim radii as is observed with younger basins[27,28]. Earlier works attributed such differences to long-term viscous flow of materials in the deep crust where temperatures were sufficiently elevated[26,29–31]. Although viscous relaxation could contribute to the muted crustal thickness signatures in the oldest basins, this process would not remove the smaller-scale topographic signatures of the crater rings at the colder surface[30].

Here we report that impact basins formed during the lunar magma ocean solidification should have produced different crater morphologies in comparison to basin morphologies forming in later epochs, comparable to example observations shown in Fig. 1.

## Results

**Effects of the melt layer on basin formation**. Numerical modelling results in Fig. 2 show the consequences of basin-forming impacts ~2-3 h after the impact, long before any long-term viscous relaxation could take effect[26,30] (see, 'Methods'). The main observation is that the morphology of the resulting impact basin is substantially different when a low-viscosity melt layer is present between the crust and mantle (right) than when it is absent (left). During impact basin formation, the transient crater typically collapses as the surrounding crust and impact-generated melt move inwards and/or upwards forming the near-final basin morphology[32]. However, when a low-viscosity melt layer is present beneath the crust (right), our impact simulations show that the basin-forming process is dominated by the inflow of the cohesionless melt layer during crater collapse. The inflow of the melt layer carries the surrounding crust towards the basin centre uninhibited by the uplift of the underlying mantle. In the upper panels of Fig. 2, the impacts produced basins similar in size to Orientale or Nectaris basins[18,33]. Figure 2 (bottom) corresponds

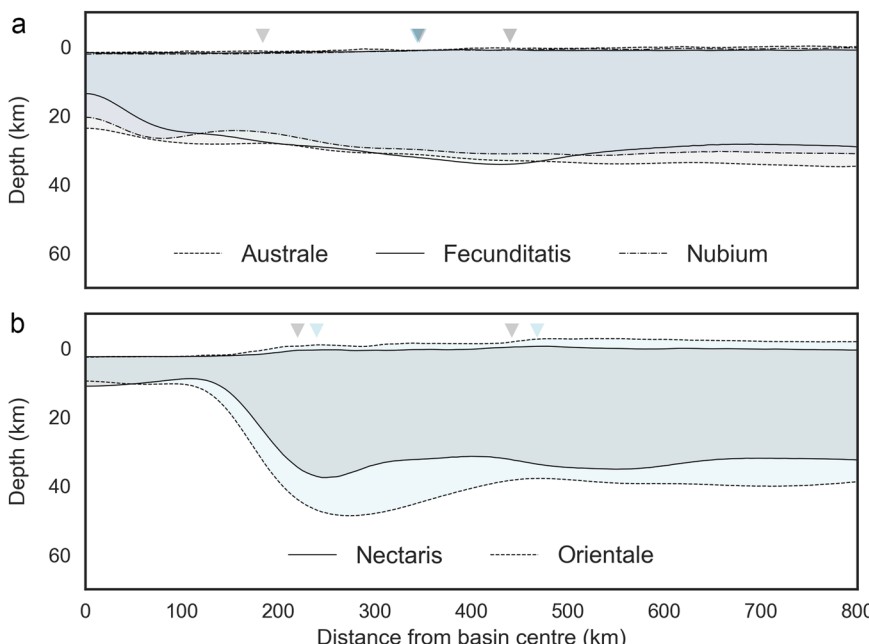

**Fig. 1 Topographic and crustal thickness profiles for lunar basins. a** Azimuthally averaged profiles of the surface relief and crust-mantle interface (derived from GRAIL gravity data[33]) for the pre-Nectarian impact basins Australe (dashed line), Nubium (dash-dot line), and Fecunditatis (solid line) compared to the likely similar sized, but younger, Nectaris (solid line) and Orientale (dashed line) basins **b**. Arrows denote previously mapped main and/or peak (inner) rim (observed or suspected)[24]. See, Supplementary Table 1 for details.

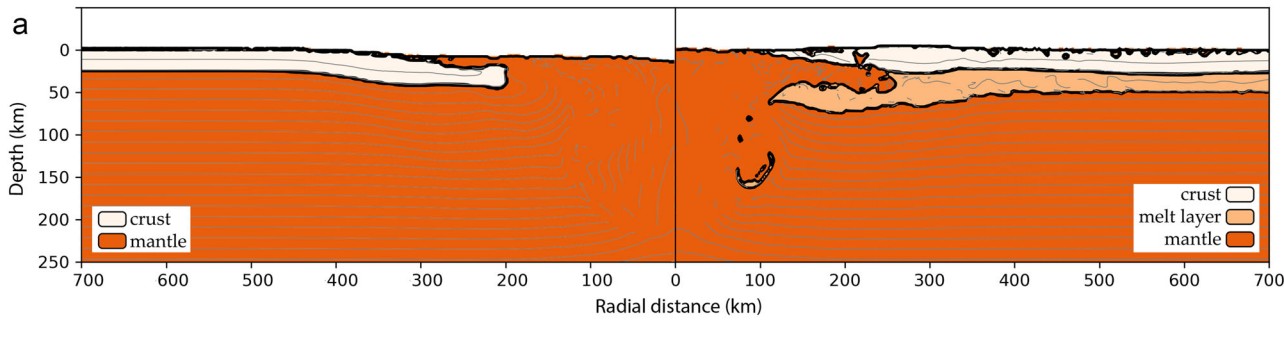

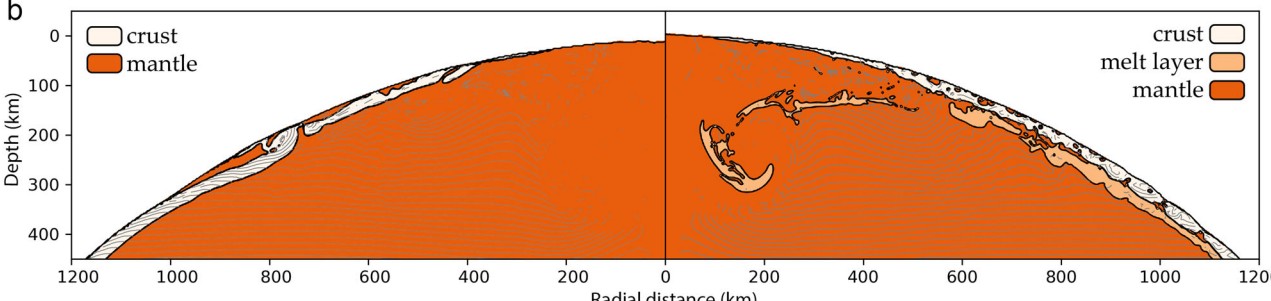

**Fig. 2 Numerical iSALE-2D simulations of impact-basin formation. a** 60-km diameter projectile striking the Moon at 17 km/s, forming an Orientale-sized basin, and **b** 120-km diameter projectile striking the Moon at 17 km/s, forming a South Pole-Aitken-sized basin. The panels on the right show results when a 25-km thick melt layer is present between a 25-km thick crust and mantle, whereas the left panels show results without a melt layer. Simulation results are shown ~3 h after impact. Results are similar for a range of impact speeds, crustal thicknesses and melt-layer thicknesses. Beige colour denotes the crust, light orange is the melt layer and dark orange is the underlying solid mantle.

to impact conditions that yield a basin similar in size to South Pole-Aitken. For both cases, the existence of a melt layer promoted crustal inflow, which then resulted in different post-impact crustal thickness signatures. These impact simulations suggest that the oldest basins should have muted crustal signatures compared to younger basins, in agreement with GRAIL observations[24]. Such an initial crustal state removes the necessity of significant crustal relaxation occurring by later long-term viscous processes. When comparing simulations with and without a melt layer, the transient crater dimensions and volume of generated impact melt remained comparable. This was to be expected given that only the rheology was modified when simulating the melt layer (from solid to low viscosity fluid), and not its temperature.

**Topographic rings formation.** Multiple rings in an impact crater form as a tectonic response of the target's lithosphere during crater formation. The number, spacing and morphology of the rings depend on the strength, temperature and thickness of the lithosphere as well as the crater size[32,34,35]. Typical lunar basins have a small number of concentric multi-ring features formed by normal faulting due to inward motion of the material filling the transient crater during the modification stage that applies stress to the lithosphere (Fig. 3 left)[24,33,36]. However, if the lithosphere is thin and/or weak (which is mimicked here by the existence of the low-viscosity melt layer), it experiences plastic failure that forms concentric graben-like structures at the surface, possibly similar to the ring structures observed on Jupiter's icy satellites Ganymede and Callisto (Fig. 3, right)[35,37].

**Crustal signature in basin formation.** Figure 4 and Supplementary Fig. 4 demonstrate that the final overall basin morphology when a melt layer is present is similar for all basin sizes, independent of the assumed initial temperature profile (Supplementary Fig. 1). The crustal thickening feature that is found near the rims of

younger basins (also known as the annular bulge)[27,28] was almost entirely removed during basin formation when as little as a 10-km-thick melt layer was present between the crust and mantle, regardless of the basin size. In contrast, for the same impact conditions without a melt layer the maximum crustal thickening near the rim can be up to two times the pre-impact crustal thickness. Furthermore, when a melt layer is present, the thickness of the crustal cap in the centre of the basin is larger than in the case where there is no melt. When a melt layer is present, the thickness of the crustal cap increases steadily from the centre outwards until it becomes comparable in thickness to the ambient crust. When no melt layer is present the crustal cap instead remained extremely thin (or, absent[38]) within the peak ring, at which point its thickness sharply increases. The same crustal morphology persists across a wide range of basin sizes (Supplementary Fig. 7), for melt layer thickness larger than 10 km (Supplementary Fig. S5), pre-impact crustal thickness between 10 km and 50 km (Supplementary Fig. S6), and for three different initial temperature profiles. Differences in basin morphologies for basins with and without a melt layer become somewhat less prominent for the largest basin, namely South Pole-Aitken size (Supplementary Figs. 6, 9), which could be due to its large size and insensitivity to the lithospheric effects on basin formation[39]. Recent work has also suggested that in the case of the South Pole-Aitken basin formation, the migration of the crust back to the basin centre after the transient crater collapse is best explained if the impact-generated melt pool had negligible viscosity[39]. The topographic relief across the basin centre showed a consistent depression for impact basins without a melt layer. However, the inner basin depression is hardly observable in the case of a basin with a melt layer, which could also be due to computational limitations.

**Discussion**

Impact bombardment has played a significant role in the evolution of the Earth-Moon system. Here we show that many ancient

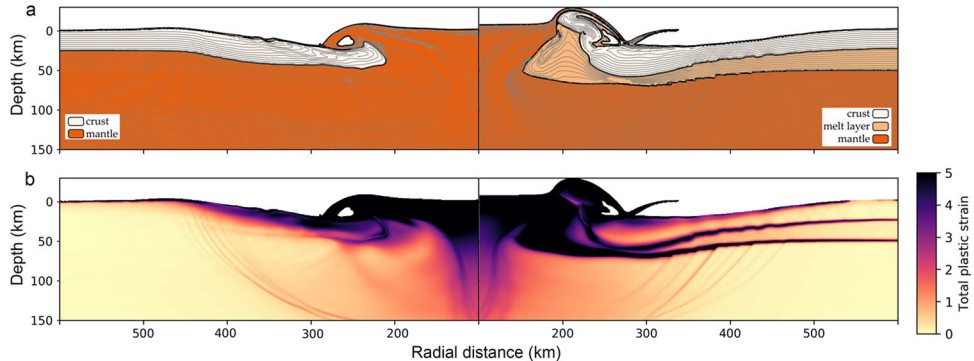

**Fig. 3 Formation of topographic rings during impact-basin formation. a** Snapshot of the basin stratigraphy and **b** total plastic strain in the late modification stage when a peak ring expected to form. The panels on the right include a melt layer between the crust and mantle, whereas the left panels show results without a melt layer. Both **a**, **b** show the formation of faults, **a** via fault slips represented by a change in stratigraphy, and **b** via localized focusing of the plastic strain. Left plots show the formation of two rings, the peak ring and the rim/outer ring (at ~270 km and 410 km), typical for the large lunar basins, and the right plots show multiple faults forming at depth that are inhibited from extending to the surface due to the existence of the melt layer (its radial range is approximated from 280 km to 460 km). Beige colour denotes the crust, light orange is the melt layer and dark orange is the underlying solid mantle. The higher the strain the darker the colours, as shown in legend.

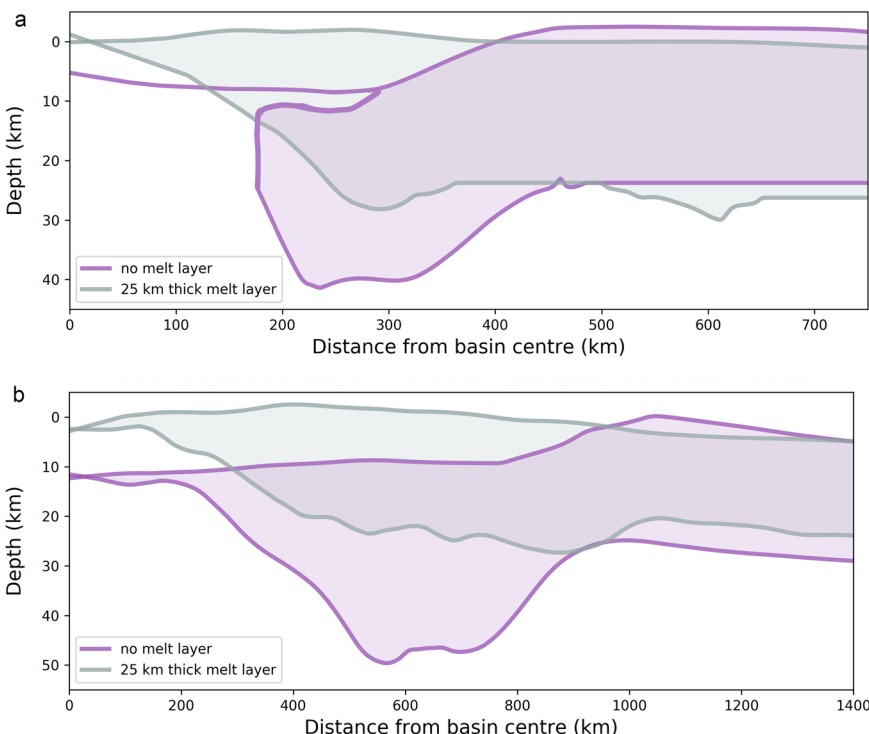

**Fig. 4 Radial profiles of the surface relief and crust-mantle interface at the end of the simulations.** Both panels showing for the case where a 25-km thick melt layer is present between the crust and mantle (grey outline) and when the melt layer is absent (purple outline). **a** impact basins forming by a 60-km diameter projectile, and **b** impact basins forming by a 120-km diameter projectile.

impact basins on the Moon, including the South Pole-Aitken basin, could have formed while the lunar magma ocean was still solidifying. Those basins would have formed with a different topographic and crustal signature in comparison to younger basins, as long as the melt layer was >10 km thick. When compared to younger basins, the crustal thickness signature would be less prominent and the topographic signature would not exhibit prominent concentric rings. In fact, the thicker the melt layer and the thinner the crust, the higher the probability that the basin would not even be recognizable in the cratering record at all, even before any long-term viscous relaxation were to take place. Thus, their number is difficult to constrain. This work is consistent with recent predictions of higher impact fluxes in the Pre-Nectarian

epoch than are inferred from the observable lunar cratering record.

## Methods

We modelled basin-forming impacts on the Moon by performing a suite of numerical impact simulations using the iSALE-2D hydrocode. iSALE is a shock physics code developed for modelling shock wave progression through geologic media[40–42] and has been validated against other hydrocodes[43]. The silicate portion of the Moon was modelled as either entirely solid or with a melt layer between the crust and the solid mantle (to mimic the residue of the magma ocean). The crust was set to either 10-, 25-, or 50-km thick, and melt layer thicknesses of 0, 10, 25 and 50 km were tested. Different layer thicknesses were assumed to mimic different stages and/or scenarios of crustal growth and late-stage magma ocean solidification. Basin morphology was found to be insensitive to the melt layer thickness once its thickness exceeded 25 km. The crust was modelled using an analytical equation of state for

granite[44] and both the melt layer and mantle were modelled using an analytical equation of state for dunite[45]. These are simplifications in terms of chemical compositions of both the crust and the melt layer, however, there is a limited number of validated and widely used constitutive models for typical rocks, which is why we used the ones that are the most similar in terms of density. The melt layer, when present, was assumed to be liquid with low constant viscosity. The melt layer was modelled simply by changing the rheology of the layer from a solid to a low-viscosity liquid, without altering the temperature of the layer. To avoid using zero values in the calculations and to mimic a near solidus melt viscosity, we adopted 100 Pa s for all simulations[46–48]. We tested viscosities of the melt layer that were several orders of magnitude higher and the crater morphology was found to remain unchanged. Two different temperature profiles were considered for most simulations; one that might be representative of the conditions immediately following magma ocean crystallization[49], and another with a temperature gradient of 50 K/km that was used in previous works when modelling the South Pole-Aitken basin[50–52]. A third temperature profile was considered in order to demonstrate that the exact properties of the chosen temperature profile have little impact on the final basin morphology. Impactors with diameters of 15, 30, 45 and 60 km were modelled as striking a flat target, whereas for larger diameters of 90, 120 and 160 km they were modelled as striking the Moon with a realistic curved surface. This was necessary because the surface curvature begins to affect the crater morphology for impact basins larger than about 1000 km[51]. These impactor sizes form basins that cover entire size range observed on the Moon, including the largest South Pole-Aitken impact basin. The impactor speed was kept constant at 10 or 17 km/s[53,54] and all impacts were modelled using an axisymmetric geometry with vertical impact conditions. The two speeds were used to cover the range of possible impact speeds, including a possible different encounter speed early in Solar System evolution[55] as well as to approximate moderately oblique impacts[56,57], because the decrease of the impact angle causes the cratering efficiency to decrease[57]. Previous studies showed that the peak ring can form slightly offset from the main ring as a consequence of a non-vertical impact angle[58], suggesting that while the ring structure can experience an offset from the centre of the structure, they still form via the same mechanisms. Therefore, as with other studies[50], we find the 2D approximation to be sufficient for the purpose of this study. Simulations were run until the crater modification stage is completed, which was confirmed by the relief of the crater's surface and crust-mantle interface reaching a stable position and not moving more than a couple of cells over a significant timestep. In real time, the equilibrium was reached within ~3 h depending on basin size. The target material model is listed in the Supplementary Notes.

## Data availability

The extended data and data analysis from numerical simulations generated in this study are provided in the Supplementary Notes. The iSALE-2D input files used to generate simulations have been deposited in the Zenodo database and accessible via https://doi.org/10.5281/zenodo.5136886.

## Code availability

This work has been produced using the iSALE shock physics hydrocode. At present, iSALE is not fully open source. Application for use of iSALE can be made via https://isale-code.github.io/. Any recent stable release can be used to reproduce the data presented.

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

## Acknowledgements
We gratefully acknowledge the developers of iSALE-2D, including Gareth Collins, Kai Wünnemann, Dirk Elbeshausen, Tom Davison, Boris Ivanov and Jay Melosh as well as the pySALEPlot visualization package (Tom Davison). K.M. acknowledges funding support from the Australian Research Council, Curtin Research Fellowship, the TIGeR institute at Curtin. Early works for this project were conducted at MIT. P.B. acknowledges funding support from the Australian Research Council.

## Author contributions
K.M. formulated the concept, conducted impact simulations, made simulation analyses and wrote up the bulk of the manuscript; M.A.W. aided in the conceptualization of the study, analysis of the impact modelling results, and writing of the manuscript; A.N. contributed with the written material about the implication of sample analyses; all other co-authors (M.L., P.A.B. and M.T.Z.) contributed to conceptualization of the study and overall writing of the manuscript.

## Competing interests
The authors declare no competing interest.
