## [Peer Review File · Nature Communications]

REVIEWER COMMENTS

Reviewer #1 (Remarks to the Author):

The key results of the paper are hydrocode models for lunar basin-forming impacts that form in the presence of a weak melt layer. The implications of these models are that the morphology of basins formed on a not-yet-fully solidified Moon are expected to be highly distinct in morphology from basins formed on a solid Moon.

The work here is straightforward, novel, and will be of broad interest for lunar and planetary scientists. I see no major issues with the manuscript. My main complaint -- if it is one -- is that a lot of the good stuff fell into the supplemental material, which perhaps an inevitable consequence of the short form. I recommend the paper be accepted following minor changes at the authors' discretion.

General comments:

-Basin topography/topographic relief. The paper clearly shows that the melt cases have really different topographic relief from the non-melt present cases (e.g. Fig 4). But the systematics of this are not described quite as clearly as I would have hoped. Does the study give enough of a handle to predict or illustrate how the post-cratering basin relief (or central depression depth) varies as a function of both size and melt-layer thickness (e.g. Fig S4/S5, recast)? A lot of the morphological differences pointed to are stratigraphic changes at tens of km scale that are very hard to observe.

-Plausible timing/melt layer properties/thickness. The uncertainty that exists about the melt layer's persistence and timing is touched on in the manuscript, but it would have been nice to have some kind of schematic that links the results presented to a cartoon for the time-evolution of the melt zone, its thickness, what it is like as it evolves. This isn't my area of expertise, but I also was wondering about the validity of presuming the layer is 100% melt, rather than interstitial partial melt. The latter still might be very low viscosity, so the qualitative signal here is presumably still valid.

-There are ideas that some of the basins that aren't recognized on the Moon might not be because of their morphology not be recognizable, but because of their erosion by repeated impacts (i.e., 'saturation') (e.g., Richardson, 2009; Richardson and Abramov, 2020). This would not obviate of the mechanism described here, but might aid in hiding early basins. I didn't see this discussed in the manuscript, so it may merit a sentence on revision.

A few specific comments on the supplemental:

Section 2.1 of supplemental: This feels like it might better be entitled "structural features in basins" or something, since the commentary is less about topography and more about faulting.

Section 2.2 of supplemental/Fig S4: it would be nice if there was an interpretation of the basin 'diameter' (or main ring) from these, along with the projectile diameters for the melt + no melt cases.

Section 2.6 of supplemental. The implications for SPA + mantle or melt sheet exposure seem really important. What exactly does a 'crustal cap' mean? A jumbled mess of pre-SPA highlands crust that flows inward?

Reviewer #2 (Remarks to the Author):

The authors address a very relevant and important questions in their numerical modelling study, namely how do large impact basins form in a solidifying magma ocean on the Moon and whether such events leave behind any detectable traces. Previous studies suggest that the very early bombardment history after the formation of the Moon is lacking in the present-day crater record suggesting that the impactor flux in the inner solar system was more intense than previously assumed. The authors show by numerical modelling

that basin formation in a partially solidified magma ocean result in different structural signatures than those characteristic for basins that formed in a completely solid crust and mantle. To my knowledge this is the first systematic study addressing this problem and, therefore, the authors results and claims are novel and relevant for a broad community dealing with the early evolution of planets and the late accretion phase. However, I have a couple of major points of criticism that require further explanation:

1. I am not fully convinced that the methodological approach is adequate and appropriate to support the drawn conclusions.

First, the partially molten magma ocean is approximated by a 10-50 km thick layer with extremely low viscosity sandwiched by a solid crust and solid mantle. The low-viscosity rheology is justified by the assumption that this layer is partially or completely molten. However, this does not correlate with the temperature profiles the authors assume at the time of impact. Apparently, the authors do not change the temperature as a function of depth such that melting occurs at the given depth range, but introduce a layer that is composed of some composition that is not specified with a low-viscosity. This simplified assumption is not well explained in the method section in supplemental material and I also question whether this approach is valid. Partial melting at some depth can be justified by high temperatures or different chemical composition. Apart from the fact that this may affect the crater formation process itself it certainly has consequences on the abundance and provenance of impact induced melting. I would expect a substantial increase in melt production, which in turn almost certainly would also affect late stage crater formation and modification processes. I think, the simplified approach this study is based on requires further justification and explanation. Maybe, additional models are required to demonstrate the applicability of the simplified setup.

Second, I noticed a few additional problems regarding the methodological approach. The authors do not provide a detailed list of model parameters and refer to previous work. In these studies, the concept of acoustic fluidization was applied. Apart from the fact that it is highly questionable whether acoustic fluidization is applicable on the given scale, the resulting crater morphology is highly sensitive to the choice of model parameters. It would be helpful to support the drawn conclusions by some sort of a sensitivity study to show how results vary for different choices of acoustic fluidization parameters.

Another problem is that the authors stop all simulations after 2-3 hours model time. It is a common and well-known problem that in models crater formation does not come to a complete stop and some target oscillations and low velocity material movements remain ongoing even after 2-3 hours model time. It is difficult to decide to what extent such late stage modifications are physical or numerical artefacts, but it should be demonstrated whether the results are converging, e.g. does the final basin depth approach some limit or so.

One last point regarding the model approach, why do the authors assume an impact velocity of 17 km/s? It is true that this value corresponds to the mean impact speed on the Moon, but for oblique impacts. In many previous studies the most likely impact conditions (17 km/s and 45°) are approximated by 2D simulations of vertical impacts using an impact speed that corresponds to the vertical component of the most likely velocity vector. This may be a minor point, but should be explained.

2. In addition to the issues I raised regarding the methodological approach I am also not entirely convinced by the interpretation of the model results. The main outcome is that no annular bulge occurs in the models with the viscous layer. This is certainly an interesting finding and better matches crustal thickness models that have been proposed for old basin structures (Fig. 1 top). However, these models have been derived from gravity data and cannot be understood to be unique. In fact, the shown crustal thickness models of Orientale and Nectaris (Fig. 1 bottom) are very different to the formation models shown in Fig. 2 (left). To better judge, what the gravity signature of the formation model would look like I suggest to derive the gravity data from the given mass distribution for a direct comparison with observational data. I presume that the density of the viscous layer may be key, which relates this point to the issue I raised above regarding the justification for the existence of a viscous layer due to higher temperatures or difference in

chemistry.

How subsequent cooling and isostatic adjustment affect the gravity signature is very speculative and requires in my opinion further discussion.

Apart from these major points, I list a few minor things that could be more easily addressed:

L10: Replace "impact simulations" by "simulations of basin formation"

L28: The formation of the Moon as a consequence of giant impact is generally accepted, but many details are highly debated. Not all scenarios would result in the formation of a LMO. I suggest to phrase more carefully. The only reference here are somewhat old and should be updated.

33: Maurice et al. (2020) suggest a younger age of the Moon: 4.425 Ga.

75: What exactly is meant by "relaxed crustal state"?

82-84: Formation of rings is not exactly understood. References here are really old. This should be phrased more carefully.

91: What exactly does Fig. 4 show? Does the line mark the boundary between the crust and the viscous layer?

Supplementary:

Fig. S1 should contain the solidus and liquidus. I suspect, that the assumed temperature profiles do not justify the existence of a melt layer. So, the solidus for chemically different materials may explain this.

Reviewer #3 (Remarks to the Author):

I found this to be a very interesting paper, with some potentially major consequences for how the planetary community should interpret the Moon's early bombardment history. The numerical simulations presented seem plausible and reasonable, though I have no particular expertise to evaluate their specifics. The assembled team knows their craft, though, and I do not expect them to have issues with their ISALE runs. Overall, I think this manuscript should be published.

With that said, though, there are some implications in the paper that warrant further discussion.

The theme of the paper is that earliest lunar bombardment record was erased, with the observed record only beginning when the magma ocean had closed/solidified some 200 Myr after the formation of the Moon. The paper also references the possibility that 200 additional basins could have formed on the Moon during the early time, roughly 5 times the observed value (e.g., Morbidelli et al 2018).

Accordingly, given that the Moon has one 2000 km basin (i.e., South Pole-Aitken basin), and its age presumably is post-magma ocean (< 4.3 Ga), the bombardment decay curve used by the referenced papers (e.g., Morbidelli et al. 2018; Zhu et al. 2019) implies the Moon had several SPA events in its early history.

However, the simulations and text seem to suggest that the melt layer has a more limited effect on the formation of such enormous basins like SPA. For example, lines 104-106:

"Differences in basin morphologies for basins with and without a melt layer become somewhat less prominent for the largest basin, namely South Pole-Aitken size (Fig. S4e), which could be due to its large size and insensitivity to the lithospheric effects on basin formation."

I do not have the trained eye to evaluate the figures, but from this text, it seems like erasing early SPA-size basins is difficult.

My request is that the authors address this issue in the paper, namely can one erase multiple SPA-size events on the surface of the Moon during the magma ocean phase. The erasure would need be extensive:

- There cannot be a topographic or gravity signature from the event that GRAIL or LRO could detect on the Moon, particularly on the farside, which has a more extensive record of ancient basins and a thicker crust.
- The impact cannot dredge up unusual interior materials from depth that could be seen as an obvious compositional anomaly on the Moon (which would presumably have been detected by M³).

If the answer is erasure can work for multiple early SPAs, it means recorded topographic/gravity history on the Moon probably starts ~ 4.3 Ga, with most early lunar history gone for good. That is an interesting prediction that has big implications for future work and for sample return mission from the Moon.

If the answer is no (or probably not), that sets up other questions, namely how many large bodies were in the bombardment population, are we using the correct decay rate for this population, and did the magma ocean really close at 4.35 Ga?

An additional possibility is that SPA formed when the magma ocean was taking place, and it survived while smaller basins faded away. If this is the case, the authors should talk about it, but I am skeptical this is a plausible solution, at least for the referenced bombardment rates. If the Moon experienced 5 times as many basins as we see now, it seems likely SPA would have experienced far more damage to its rim (and to its gravity signature) than we see now.

Otherwise, I have a few minor comments:

Line 33. I do not really believe it, but there are a number of fairly recent papers out there suggesting the Moon formed at the relatively young age of ~ 4.4 Ga. Should this be mentioned and/or referenced?

Note that if the Moon formed at ~ 4.4 Ga, the issue of the missing early basins goes away. That does not mean the simulations in the paper are incorrect, only that they might be moot.

Lines 38-42. My recollection is that with few exceptions, the ancient lunar zircons only go back to $\sim 4.2-4.3$ Ga. The shock deformation events have to be younger than these zircon formation ages, and that would place them outside the first 200 Myr of lunar history.

Best regards,
Bill Bottke

Response to the reviews:

Reviewer #1 (Remarks to the Author):

“... I recommend the paper be accepted following minor changes at the authors' discretion.”

Thank you!

General comments:

-Basin topography / topographic relief. The paper clearly shows that the melt cases have really different topographic relief from the non-melt present cases (e.g. Fig 4). But the systematics of this are not described quite as clearly as I would have hoped. Does the study give enough of a handle to predict or illustrate how the post-cratering basin relief (or central depression depth) varies as a function of both size and melt-layer thickness (e.g. Fig S4/S5, recast)? A lot of the morphological differences pointed to are stratigraphic changes at tens of km scale that are very hard to observe.

We would first like to emphasize that the majority of our interpretation is based on the crustal thickness profiles and ring spacing, which are relatively easy to characterize in our simulations. As the reviewer notes, characterizing the topographic profile is more difficult. This is in part because the relative variations in surface relief are small with respect to the resolution of the model, which was at least a few km, depending on the basin size.

We added the following explanation to SI: “In this work, we focus on the formation of basin rings, their spacing and the final crustal thickness variations. Though the final surface relief is also an important outcome of basin formation, this was not easy to interpret for many of our iSALE simulations. For example, for the same size impact basin, when the basin formation was modelled until completion, the cell dimension was 2.5 km by 2.5 km, but we used 500 by 500 m cells when simulating formation of faults. Many of our highest resolution simulations that focused on faulting did not run to completion, due to high resolution that would extend the run times to weeks in real time. In simulations where the melt layer was present, even after 3 h following basin formation, there were vertical oscillations (equal to a couple of cells moving up-down) that affected the entire numerical mesh, so both the crust and upper mantle and not the crustal thickness. For this reason, we do not interpret the final surface relief predicted by our models in this work, but the resulting crustal thickness profile. We note that in this study, we made between 150 and 200 simulations all of which required day to weeks-long runtimes.”

With this in mind, we added the following to the main text: “The topographic relief across the basin centre suggested a consistent depression for impact basins without a melt layer. However, the inner basin depression is hardly observable in the case of a basin with a melt layer, which could also be due to computational limitations.”

-Plausible timing / melt layer properties / thickness. The uncertainty that exists about the melt layer's persistence and timing is touched on in the manuscript, but it would have been nice to have some kind of schematic that links the results presented to a cartoon for the time-evolution of the melt zone, its thickness, what it is like as it evolves. This isn't my area of expertise, but I also was wondering about the validity of presuming the layer is 100% melt, rather than interstitial partial melt. The latter

still might be very low viscosity, so the qualitative signal here is presumably still valid.

In the original manuscript, we stated “Recent studies suggest that lunar magma ocean (LMO) solidification could have taken up to ~200 Myrs^{6,12-14}, and this suggests that a significant portion of basin forming impacts could have occurred while the LMO was still solidifying.” And “The timeframe for the solidification of the lunar magma ocean (LMO) varies significantly between calculations^{22,23}, from within a few Myr²⁴ to up to ~200 Myr^{6,12-14,25}, but could also have varied regionally for up to ~500 Myr^{26,27} Radiogenic lunar crustal ages span from 4.47 Ga to 4.31 Ga, which falls broadly within this range, and the age of the giant impact has been estimated to have occurred at ~4.54-4.425 Ga^{23,24,28}.”

We feel that this provides more than enough information about the time evolution of the magma ocean: the time evolution of the melt layer thickness is highly uncertain. It is for this reason that we have run several simulations testing the sensitivity of the melt layer thickness. As described in the original text, our conclusions are unmodified for all thicknesses greater than 25 km. The reviewer makes a good point that we don't know whether the “melt” layer is 100% molten, or if it might be some kind of partially solidified “mush”.

The effective viscosity of partially molten materials will be significantly less than for entirely molten materials. Solomatov (2007) suggests that the typical viscosity of near-liquidus materials in a magma ocean is 0.01 Pa s with a factor of 10 uncertainty. We did test varying by several orders of magnitude the viscosity of the melt layer, and this had no effect as long as the melt layer viscosity was less than approximately 1e10 Pas, which is significantly lower than the solid portion of the crust and mantle. With this range in mind, our results are more appropriate for layer that is significantly molten, which was the aim of this work.

Solomatov, V. S., Magma oceans and primordial mantle differentiation, in *Treatise on Geophysics*, edited by G. Schubert, Elsevier, v. 9, pp. 91-120, 2007.

We added this text to the numerical setup section in SI: “Solomatov (2007)¹⁰ suggests that a typical viscosity of near-liquidus materials in a magma ocean is 0.01 Pa s with a factor of 10 uncertainty. Our tests showed that the chosen viscosity of the melt layer had little influence on the final basin morphology for all values less than about 1010 Pa s, which is significantly lower than the viscosity of a solid rock. In this work, we used a constant viscosity of 100 Pa s, simply to avoid using (near) zero values in calculation.” We also added small clarifications in Section 1 and around Figs S1 and S3 relating to the target temperature and yield strength profiles with depth.

-There are ideas that some of the basins that aren't recognized on the Moon might not be because of their morphology not be recognizable, but because of their erosion by repeated impacts (i.e., 'saturation') (e.g., Richardson, 2009; Richardson and Abramov, 2020). This would not obviate of the mechanism described here, but might aid in hiding early basins. I didn't see this discussed in the manuscript, so it may merit a sentence on revision.

The reviewer makes a good point. In the original manuscript, we noted that basins could be degraded either “at birth” by having a low viscosity layer beneath the crust,

or by long-term viscous relaxation of the crust. The reviewer notes that these basins could also be degraded by subsequent “impact erosion.” Impact degradation of these basins, however, should only affect the surface topography, and not the crustal thickness signature. In the revised text, we have noted this and reworded some of the text:

“The topographic signatures of the oldest basins could have been degraded by subsequent impact bombardment^{32,33}. Gravity data acquired by the Gravity Recovery and Interior Laboratory (GRAIL) mission³⁴, however, have shown that the large and stratigraphically oldest pre-Nectarian impact basins have muted subsurface crustal signatures compared to the younger Nectarian and Imbrian impact basins^{35,36}”

A few specific comments on the supplemental:

Section 2.1 of supplemental: This feels like it might better be entitled "structural features in basins" or something, since the commentary is less about topography and more about faulting.

We changed the title of S2.1 to “Structural features (ring formation) in basins”

Section 2.2 of supplemental / Fig S4: it would be nice if there was an interpretation of the basin 'diameter' (or main ring) from these, along with the projectile diameters for the melt + no melt cases.

We note that the projectile diameters are the same in the case of melt / no melt, for each panel. The dependence of basin structure on projectile diameter is well known (e.g., Miljkovic et al., 2013; 2016) for the case where a melt layer is not present. However, the final diameter was hard to measure for the case where there was a melt layer, which is why we did not report a number in the text. Nevertheless, the basin diameters for simulations without a melt layer have been estimated and we now note in the caption of Figure S4: *“The final crater diameter (based on the location of the inner rings and crustal thinning) for basins without the melt layer are approximately a) 300 km, b) 500 km, c) 700 km, d) 1000 km, and e) 1600 km. The uncertainty in these diameter estimates are about 10 km for the smallest crater, and up to 100 km for the largest. Diameters are not estimated for the case where a melt layer is present (right) given the lack of clearly identifiable crustal thickness characteristics and the multitude of faults in the crust.”*

Section 2.6 of supplemental. The implications for SPA + mantle or melt sheet exposure seem really important. What exactly does a 'crustal cap' mean? A jumbled mess of pre-SPA highlands crust that flows inward?

The following text was added to the section about SPA: *“Our simulations suggest that the crustal inflow is composed mostly of broken rafts flowing back into the basin centre that are composed primarily of overturned and jumbled crust originating from lower and mid-level crust levels.”*

Reviewer #2 (Remarks to the Author):

1. I am not fully convinced that the methodological approach is adequate and appropriate to support the drawn conclusions. First, the partially molten magma ocean is approximated by a 10-50 km thick layer with extremely low viscosity

sandwiched by a solid crust and solid mantle. The low-viscosity rheology is justified by the assumption that this layer is partially or completely molten. However, this does not correlate with the temperature profiles the authors assume at the time of impact. Apparently, the authors do not change the temperature as a function of depth such that melting occurs at the given depth range, but introduce a layer that is composed of some composition that is not specified with a low-viscosity. This simplified assumption is not well explained in the method section in supplemental material and I also question whether this approach is valid. Partial melting at some depth can be justified by high temperatures or different chemical composition. Apart from the fact that this may affect the crater formation process itself it certainly has consequences on the abundance and provenance of impact induced melting. I would expect a substantial increase in melt production, which in turn almost certainly would also affect late stage crater formation and modification processes. I think, the simplified approach this study is based on requires further justification and explanation. Maybe, additional models are required to demonstrate the applicability of the simplified setup.

The reviewer is correct that our assumed temperature profiles are not entirely self-consistent with the melt layer that is present beneath the crust. In our study, we tried to see what would be the consequence of adding a melt layer at the base of the crust *if all other variables were equal*. For our nominal case, the chosen temperature profiles do not predict the existence of melt at the time of the impact. To investigate how a melt layer would affect the basin morphology, we simply changed the viscosity of the material to a value that is appropriate of molten materials.

This could be criticized as not being self-consistent with the chosen temperature profile. But as the reviewer acknowledges, we don't know a priori the composition of the melt layer, and hence we don't know its liquidus and solidus temperatures. Also, it is not clear how one would compare two simulations, one with and one without a melt layer, when the initial temperature profile is different. If the reviewer has a suggestion for a specific change in this regard, we would be happy to implement it.

In any case, we agree with the reviewer that the methodology of simply changing the viscosity of melt layer (regardless of the temperature profile and unknown solidus and liquidus temperatures) was not entirely clear. We have added the following text that describes this in more detail, and which notes the motivation for simply changing one variable in our simulations.

"We note that our simulations with a melt layer are not entirely self-consistent with the temperature profiles in Figure S1. Our approach was to investigate the consequence of adding a melt layer at the base of the crust while keeping all other variables constant. Given that the composition of the melt layer is uncertain, estimating its liquidus and solidus temperatures would also be uncertain. Furthermore, the composition of the melt layer changes as the magma ocean continues to crystallize. To investigate how a melt layer would affect the basin morphology, we thus simply changed the viscosity of the material to a low, non-zero, value (100 Pa s), that is appropriate of molten materials within magma oceans¹⁰ while leaving the temperature of the melt unchanged.

Second, I noticed a few additional problems regarding the methodological approach. The authors do not provide a detailed list of model parameters and refer to previous

work. In these studies, the concept of acoustic fluidization was applied. Apart from the fact that it is highly questionable whether acoustic fluidization is applicable on the given scale, the resulting crater morphology is highly sensitive to the choice of model parameters. It would be helpful to support the drawn conclusions by some sort of a sensitivity study to show how results vary for different choices of acoustic fluidization parameters.

We thank the reviewer for noting that some model parameters were missing from the SI. Though these are found in the appropriate references we provided, we have decided to create a new table (Table S2) listing all parameters for the benefit of the reader.

We added the following text in S1 section: “The list of input parameters for our iSALE simulations is shown in Table S2. These models are very similar to the models used extensively in previous lunar basin modelling^{1,11–18}.”

We appreciate the comment concerning acoustic fluidisation, but this has been extensively studied in previous works since its conception by Melosh in 1970s. For impact basin formation the optimal parameters were identified such that the final basin morphology is comparable to observations. We are simply using the best state of the art values that are widely used in the iSALE community and were applied in the last 10 years of publications on this topic (e.g., Ivanov et al., 2010, Potter et al. 2012, 2015, Johnson et al. 2016, Zhu et al., 2015, Lompa et al. 2021, Miljkovic et al. 2013, 2016), etc.

Specifically, sensitivity to the material model, especially to acoustic fluidization, has been extensively studied in our past publications related to lunar basin formation. Miljkovic et al. 2013 showed that basins forming in warm/hot gradients do not need to have acoustic fluidisation applied (final basin morphology forms the same in case when the acoustic fluidisation is and isn't included), whereas basins forming in a cold gradient need to have it included in the model. For cooler temperature gradients, the final basin morphology forms a basin that is too deep compared to observations. Here, we applied acoustic fluidisation parameters that we used in our previous works (Miljkovic et al., 2013, 2015, 2016, 2017) for both cases (with and without melt) because our thermal gradient here is comparable to cool cases in Miljkovic et al. previous works.

We added the following to the SI: “Previous studies investigated the effects of acoustic fluidisation on large crater formation and found that the inclusion of acoustic fluidisation had little to no influence on the final basin morphology. In particular, the basin depth was largely unchanged when using acoustic fluidization with warm and hot temperature profiles in the target¹⁹. However, studies have shown that it was necessary to include acoustic fluidisation in order to match the observed depth of the basin floor when using colder temperature profiles¹¹. Acoustic fluidisation is included in this work because the temperature profiles considered here are comparable to the cold profiles used in Miljkovic et al.¹¹. Acoustic fluidisation was applied in the same way on targets both with and without a melt layer. “

Another problem is that the authors stop all simulations after 2-3 hours model time. It is a common and well-known problem that in models crater formation does not come to a complete stop and some target oscillations and low velocity material

movements remain ongoing even after 2-3 hours model time. It is difficult to decide to what extent such late stage modifications are physical or numerical artefacts, but it should be demonstrated whether the results are converging, e.g. does the final basin depth approach some limit or so.

The crater formation never comes to a complete stop. This is the numerical side effect of the code known to the iSALE community. However, it is important to note that these oscillations are very small (within a couple of cells vertically or horizontally) and have no physical meaning. In our simulations, very little horizontal movement occurs after 3 h of model time between recorded timesteps. More importantly, faulting occurs during the first 2 hours (before the final morphology is reached). Thus, the crustal thickness signature and location of faults all form before we stop our simulations. It is true that these oscillations, however, could affect the final topographic profile of the basin, but our work mostly concerns the large lateral change in crustal thickness that are not affected by these numerical oscillations.

We added this explanation into the methodology section: "Simulations were run until the crater modification stage is completed, which was confirmed by the relief of the crater's surface and crust-mantle interface reaching a stable position and not moving more than a couple of cells over a significant timestep. In real time, the equilibrium was reached within ~3 h depending on basin size."

One last point regarding the model approach, why do the authors assume an impact velocity of 17 km/s? It is true that this value corresponds to the mean impact speed on the Moon, but for oblique impacts. In many previous studies the most likely impact conditions (17 km/s and 45°) are approximated by 2D simulations of vertical impacts using an impact speed that corresponds to the vertical component of the most likely velocity vector. This may be a minor point, but should be explained.

This has already been included in the methods section in the main text: "The impactor speed was kept constant at 10 or 17 km/s¹³ and all impacts were modelled using an axisymmetric geometry with vertical impact conditions. The two speeds were used to cover the range of possible impact speeds, including a possible different encounter speed early in Solar System evolution¹⁴ as well as to be a proxy for a moderately oblique impacts¹⁵⁻¹⁶, because the decrease of the impact angle causes the cratering efficiency to decrease¹⁶."

As described in the Methods section, we considered both 10 km/s and 17 km/s. However, the choice of speeds has not shown to have a significant effect on the final crater morphology, which is why it wasn't extensively shown here.

We add another sentence to the supplemental material to address this: "In our simulations, we used both 17 km/s and 10 km/s for the bolide impact speed. The two speeds did not show significant differences in basin morphologies when the outcomes are analyzed in terms of the kinetic energies of the impactor. We chose to focus on the 17 km/s impact speed in order to limit the number of free parameters in our simulations."

2. In addition to the issues I raised regarding the methodological approach I am also not entirely convinced by the interpretation of the model results. The main outcome

is that no annular bulge occurs in the models with the viscous layer. This is certainly an interesting finding and better matches crustal thickness models that have been proposed for old basin structures (Fig. 1 top). However, these models have been derived from gravity data and cannot be understood to be unique. In fact, the shown crustal thickness models of Orientale and Nectaris (Fig. 1 bottom) are very different to the formation models shown in Fig. 2 (left). To better judge, what the gravity signature of the formation model would look like I suggest to derive the gravity data from the given mass distribution for a direct comparison with observational data. I presume that the density of the viscous layer may be key, which relates this point to the issue I raised above regarding the justification for the existence of a viscous layer due to higher temperatures or difference in chemistry. How subsequent cooling and isostatic adjustment affect the gravity signature is very speculative and requires in my opinion further discussion.

The reviewer first notes that the crustal thickness models derived from gravity data are not “unique”. Though this is in principle true, if you assume a constant density for the crust and mantle (along with an average crustal thickness constrained by seismic data), the crustal thickness inversion is then in fact unique. The density of the mantle is fairly well constrained from petrological arguments and moment of inertia analyses, and the crustal density has been estimated using high-resolution GRAIL gravity data (e.g., Wieczorek et al. 2013).

The reviewer then suggests that instead of comparing the predicted iSALE crustal thickness profiles with those from the gravity inversions, that we should instead compute synthetic gravity and compare with the observed gravity field. There are two problems in doing this. First, just as with the GRAIL-crustal thickness models, you would need to assumed densities (and porosities) of the crustal and mantle materials. Choosing these values are largely non-unique. Secondly, as the reviewer notes above, at 3 hours of simulation time, there are still some vertical oscillations that occur. These vertical oscillations give rise to significant variations in the computed gravity field! The reviewer seems to acknowledge these problems by stating “How subsequent cooling and isostatic adjustment affect the gravity signature is very speculative” which is why we tried to avoid this problem. The easiest approach is thus to compare crustal thickness, and not gravity.

We added the following in the SI: “In our work, we compare the relief of the surface and crust-mantle interface with the known surface topography of the Moon and GRAIL-derived crustal thickness models. An alternative approach could have been instead to compare directly the gravity field predicted by iSALE with GRAIL observations. However, to do so, it would have been necessary to specify the density and porosity of the crust, which are both uncertain. Any numerical oscillations in the vertical direction at 2-3 h time in the iSALE simulations would also have affected the gravity signal. Furthermore, we note that isostatic adjustment of the basin (which is largely vertical in nature) would modify the predicted gravity field but would not affect significantly the crustal thickness. For these reasons, we chose to the analyse the crustal thickness profiles instead of the predicted gravity.”

Apart from these major points, I list a few minor things that could be more easily addressed:

L10: Replace “impact simulations” by “simulations of basin formation”
Changed to “impact basin-forming simulations”

L28: The formation of the Moon as a consequence of giant impact is generally accepted, but many details are highly debated. Not all scenarios would result in the formation of a LMO. I suggest to phrase more carefully. The only reference here are somewhat old and should be updated.

We edited L28 statement: “The long-standing Moon-formation theory whereby a giant impact occurred with proto-Earth implies that the young Moon formed with a global magma ocean^{12–14,21–24}” and we added additional more recent references.

33: Maurice et al. (2020) suggest a younger age of the Moon: 4.425 Ga.

We edited the following statement in the same line to: “giant impact has been estimated to have occurred at ~4.54-4.425 Ga [12, 23,24,28]”, Ref 12 is Maurice et al., 2020.

75: What exactly is meant by “relaxed crustal state”?

We understand that this sentence was misleading. We made the following edit: “These impact simulations suggest that the oldest basins should have muted crustal signatures compared to younger basins, in agreement with GRAIL observations³⁶. Such an initial crustal state removes the necessity of significant crustal relaxation occurring by later long-term viscous processes.”

82-84: Formation of rings is not exactly understood. References here are really old. This should be phrased more carefully.

We supplied both old and recent references for ring formation: Melosh and McKinnon (1978), McKinnon (1981) who explained the faulting mechanism using basin geophysical interpretations and more recent Johnson et al. (2016; 2018) that confirmed the old theoretical explanation using numerical impact simulations.

91: What exactly does Fig. 4 show? Does the line mark the boundary between the crust and the viscous layer?

Yes. The caption of Figure 4 says “Radial profiles of the lunar crust...”

Supplementary:

Fig. S1 should contain the solidus and liquidus. I suspect, that the assumed temperature profiles do not justify the existence of a melt layer. So, the solidus for chemically different materials may explain this.

We added the following explanation to caption of Figure S1: “Neither profile has temperatures above the material’s solidus.”, as already stated above and explain how (and why) temperature was treated in our modelling.

Reviewer #3 (Remarks to the Author):

I found this to be a very interesting paper, with some potentially major consequences for how the planetary community should interpret the Moon’s early bombardment history. The numerical simulations presented seem plausible and

reasonable, though I have no particular expertise to evaluate their specifics. The assembled team knows their craft, though, and I do not expect them to have issues with their ISALE runs. Overall, I think this manuscript should be published.

The theme of the paper is that earliest lunar bombardment record was erased, with the observed record only beginning when the magma ocean had closed/solidified some 200 Myr after the formation of the Moon. The paper also references the possibility that 200 additional basins could have formed on the Moon during the early time, roughly 5 times the observed value (e.g., Morbidelli et al 2018). Accordingly, given that the Moon has one 2000 km basin (i.e., South Pole-Aitken basin), and its age presumably is post-magma ocean (< 4.3 Ga), the bombardment decay curve used by the referenced papers (e.g., Morbidelli et al. 2018; Zhu et al. 2019) implies the Moon had several SPA events in its early history.

However, the simulations and text seem to suggest that the melt layer has a more limited effect on the formation of such enormous basins like SPA. For example, lines 104-106:

“Differences in basin morphologies for basins with and without a melt layer become somewhat less prominent for the largest basin, namely South Pole-Aitken size (Fig. S4e), which could be due to its large size and insensitivity to the lithospheric effects on basin formation.”

I do not have the trained eye to evaluate the figures, but from this text, it seems like erasing early SPA-size basins is difficult.

The possibility that there could have been several SPA-sized events in lunar history is an interesting one. However, when dealing with a single basin, you really need to worry about small number statistics! It is possible that the sole SPA event was a statistical outlier, and does not represent that average number of SPA-sized basins that should have formed. Thus, multiplying 1 by 5 might not be the correct statistical average for how many of these basins should have formed.

My request is that the authors address this issue in the paper, namely can one erase multiple SPA-size events on the surface of the Moon during the magma ocean phase. The erasure would need be extensive:

- There cannot be a topographic or gravity signature from the event that GRAIL or LRO could detect on the Moon, particularly on the farside, which has a more extensive record of ancient basins and a thicker crust.
- The impact cannot dredge up unusual interior materials from depth that could be seen as an obvious compositional anomaly on the Moon (which would presumably have been detected by M³).

If the answer is erasure can work for multiple early SPAs, it means recorded topographic/gravity history on the Moon probably starts ~4.3 Ga, with most early lunar history gone for good. That is an interesting prediction that has big implications for future work and for sample return mission from the Moon.

If the answer is no (or probably not), that sets up other questions, namely how many large bodies were in the bombardment population, are we using the correct decay rate for this population, and did the magma ocean really close at 4.35 Ga?

An additional possibility is that SPA formed when the magma ocean was taking place, and it survived while smaller basins faded away. If this is the case, the authors should talk about it, but I am skeptical this is a plausible solution, at least for the referenced bombardment rates. If the Moon experienced 5 times as many basins as we see now, it seems likely SPA would have experienced far more damage to its rim (and to its gravity signature) than we see now.

In addition to clarifications throughout the main text and the SI regarding what we could model in this work and why, we would like to further clarify the aspects of basin morphologies observed. We added the following discussion at the end of the SM: “The smaller basins tend not to form the crustal thickening surrounding the crustal thinning, but instead show a more gradual crustal thickness profile. For basins that are as large as the South Pole-Aitken basin, the effects of including a melt layer are not as prominent when compared to the simulation results of smaller basins (namely in terms of crustal inflow and extent of the crustal thinning).” It was difficult to talk about complete erasure here other than the relative difference in impact basin morphologies between smaller ones and the SPA-sized ones in the context of their size and overall morphology vs existence of melt.

Otherwise, I have a few minor comments:

Line 33. I do not really believe it, but there are a number of fairly recent papers out there suggesting the Moon formed at the relatively young age of ~4.4 Ga. Should this be mentioned and/or referenced?

As also suggested by Reviewer 2, we added a reference to Maurice et al., 2020 that cites 4.425 Ga as lunar formation age.

Note that if the Moon formed at ~4.4 Ga, the issue of the missing early basins goes away. That does not mean the simulations in the paper are incorrect, only that they might be moot.

We don't think that this is necessarily true. As we stated in the main text “A recent reconstruction of the late-accretion history of the Moon based on impact-delivered siderophile elements has suggested that there could have been as many as 200 basin-forming impacts that formed before 4.35 Ga that are unaccounted for in the current lunar cratering record”. In this regard, it doesn't matter if the Moon formed at 4.36 Ga or 4.5 Ga: There is simply evidence that more basins formed than are visible in the cratering record.”

Lines 38-42. My recollection is that with few exceptions, the ancient lunar zircons only go back to ~4.2-4.3 Ga. The shock deformation events have to be younger than these zircon formation ages, and that would place them outside the first 200 Myr of lunar history.

If the Moon formed later (Maurice et al., 2020) then this would make sense. We don't give exact zircon ages, but refer to the oldest of them that were recorded (e.g., Crow et al., 2017 or Nemchin et al. 2010 that were already referenced).

Some small grammar updates were marked in blue color.

REVIEWER COMMENTS

Reviewer #1 (Remarks to the Author):

I have no remaining issues with the manuscript and recommend its publication.

Reviewer #2 (Remarks to the Author):

I very much appreciate the detailed response to my comments on the manuscript. The authors provide well thought-through arguments to my points of criticism, but I am still not fully convinced that the methodological approach is appropriate. The authors do not provide any additional data that could prove the applicability of their method. Although they now clarify in the supplementary material that the remnants of a magma ocean are approximated by a liquid layer the main text still infers that a layer of molten mantle material was modelled. In addition, my comment on the comparison between the gravity-derived crustal thickness model and the basin formation model was not addressed satisfactorily in their rebuttal.

 (1.1) The authors clarify the setup and state that their approach is thermodynamically not self-consistent in the supplementary material, but not in the main text. I think, this important information should be mention in the method part of the main text as I consider it as a critical simplification. In addition, I suggestion the replace "melt layer" by "liquid" or "viscous layer" all through the text to avoid any misunderstanding.

I agree, that a thermodynamically consistent setup is challenging. There are many unknowns making it difficult to better constrain a more realistic setup. However, it is a major simplification and I would expect here some discussion whether this rather simple approach is justified or not. The impact process itself generates a substantial amount of melt that would mix with the pre-impact melt. In my opinion, it cannot be ruled out that this has some effect. The authors claim that the "volume of impact melt generated remained comparable" (line 80), which does not surprise me with the given setup, but if a more thermodynamically consistent setup had been chosen, this would certainly change significantly. I would suggest to assume a temperature profile and solidus that generates melting in a certain depth range equivalent to the assumed liquid layer. Such a setup could be used to test whether the liquid layer approach is suitable or not. Such a test should be included in the supplementary material.

 (1.2) Generally, in my opinion, the effect of acoustic fluidization on basin formation is not well understood. Apparently, some weakening mechanism is required to reproduce the observed crater morphology, but it is difficult to justify that the mechanism of AcFI could actually work at a size-scale of basins. Well, this paper is not about AcFI and I acknowledge the text that was added to the supplementary material to justify the chosen parameters.

However, the authors write in their rebuttal "Miljkovic et al. 2013 showed that basins forming in warm/hot gradients do not need to have acoustic fluidisation applied (final basin morphology forms the same in case when the acoustic fluidisation is and isn't included), whereas basins forming in a cold gradient need to have it included in the model." I may have overlooked it, but I couldn't find anything in Miljkovic et al. (2013) showing that this assumption holds true. The reference that was added to the text, Ivanov et al. (2019), does not consider AcFI at all, if I am not mistaking. The transition from a temperature regime where AcFI is required to a regime where basin formation may work without AcFI most likely depends on the temperature profile and, thus, is related to my previous comment on the thermal state and a thermodynamic self-consistency. I think, these thinks are coupled which is why I think the matter of AcFI cannot be excluded from the discussion.

 (2) The authors did not reply to my question regarding the comparison of gravity-derived models of crustal thickness in Fig. 1 and the formation model in e.g. Fig.2 or Fig. 4. In the formation model the mantle uplift has a different shape (crustal slap or melt-layer are sandwiched in between mantle material) than in the gravity-derived model where a simple dome structure is assumed. To my understanding the crustal thickness inversion only accounts for a simple two-layer case that does not account for the fact that the

structure of the central mantle uplift may be more complicated. I agree, computing the gravity signature from the formation model is not straight forward so I cannot provide a solution here, but I would expect some discussion and further explanation how the comparison is made. The stated similarity between formation model and crustal thickness is not obvious to the reader. In fact, as I mentioned already in my previous review, the gravity signature is related to the composition of the melt layer. It is nowhere stated whether the liquid layer is considered to be part of the mantle or whether it has a composition that is rather similar to the crust. I guess, if this is some end-member product of the magma ocean crystallization I presume its density may be somewhere in between the crust and mantle. I think, this should be discussed in the main text.

Finally, a few minor points need to be revised:

In Fig 4. according to the legend the 25 km melt layer is shown, but in the caption it is stated that the melt layer was only 10 km thick

At line 100 it says "When no melt layer is present the crustal cap instead remained extremely thin within the peak ring...", this only holds true for the larger impactors. For the 60 km impactor (Fig. 4) no crustal cap occurs at all. This may require some discussion.

At line 135 the authors state that the crust was modelled "...using an analytical equation of state for granite...". This needs to be explained as the lunar crust is certainly not composed of granite. A reference may be sufficient here as it is not uncommon to use granite, but I think, for somebody not familiar with modeling of crater formation on the Moon may be confused by this.

Reviewer #4 (Remarks to the Author):

Review for Miljković et al., "Cryptic impact cratering during lunar magma ocean solidification"

The present article proposes a novel explanation for the apparent lack of very old impact basins on the Moon, relying on the possibility the signature of that craters formed during the lifetime of the lunar magma ocean isn't conserved. As these results have potential far-reaching consequences in terms of how we understand the chronology of the early Moon, and in turn, the evolution of the solar system, and since the mechanism is simple enough and the results seem robust enough, I think it justifies publication in a broad scope journal as Nature Communications.

I join at an advanced stage of the review process, so many points have already been raised by other reviewers and addressed by the authors. As a consequence, most of my concerns and comments address minor points and aim at easing the understanding to a broad audience that might not be familiar with impact modeling (myself included). While the point is clearly made and the article well written, I think that it would benefit from extending the discussion on several points. The main text is very short, so that this can be done without trespassing the journal's requirements.

My main objection is that, in spite of criticisms raised by the other reviewers, the authors haven't made clear that the initial temperature profile plays only a secondary role. Simulations have been made with two different profiles, but no comparison is shown, and the reader must be content with the simple claim that the differences are minor, relegated to the supplemental material, and without it being supported by any quantitative information. Furthermore, it is never stated which of the two profiles is employed in the results presented. Since the authors have worked extensively on characterizing the influence of the thermal profile on the crater morphology in previous studies, I expect them to have a few more relevant things to add to the discussion. In particular because the presence or absence of a molten layer likely correlates with a specific thermal state (hot profile when a molten layer is still present and cold layer when the mantle is fully solid).

I can imagine that the impact-induced melt pond in the crater is a transient feature whose rapid evolution is hard to accurately describe. In the present case, do you expect (or is the model implicitly solving for) interactions between it and the pre-existing molten layer? Following this line, although I understand the choice of representing the 3 different components (crust, molten layer and mantle) in Fig. 2, I wonder how relevant it is to distinguish between the molten layer (which is no longer much of a layer after the impact) from the molten part of the mantle (which is not distinguished from the mantle). How deep is the melt pond in the crater at this stage, and would it be possible to represent it on the Figure?

Is it possible to draw conclusion on the excavation of material from the molten layer, which could have a clear compositional signature (being composed of very evolved liquids)? For instance, in the left panel of Fig. S4a, material from the molten layer seems to be excavated at the center of the crater, but this is still a very early stage, and long-term processes not addressed here might blur this signature.

Another long-term process that would be worth mentioning in the case of large basins is impact-induced magmatism in the mantle (e.g. Elkins-Tanton et al., 2004), as it could be important if the mantle is still hot, and could strongly affect the crater signature.

Finally, being a bit more exhaustive in terms of which of the ancient craters might be concerned by this enhanced relaxation, and whether one can see a transition (and a potential link between the crater stratigraphic ages and the magma ocean lifetime), as was done for instance by Conrad et al., 2018, would be an interesting addition.

Main text

L. 50: "age criteria P-13 and P-14": I think that if this age criterion is to be mentioned, it deserves a minimum of precision (merely referring to the literature distracts the unfamiliar reader).

L. 53: "between one and two radii from the basin center": Can you explicit how the crater diameter is defined? (is it the topographic ring diameter as indicated in the caption of Fig. 1?)

L. 84: write "depend" (without "s")

L. 98-99: "Furthermore, when a melt layer is present, the thickness of the crustal cap in the centre of the basin is larger than in the case where there is no melt: It seems that for the Orientale-like impact, there is no crustal cap at all at the centre.

Fig 2. For the SPA-like case without liquid layer, there seems to be a significant layer of mantle ejecta covering the crust as far as >700 km (even more with the curvature) from the center, while the crust remains exposed in the case with a liquid layer. How does it compare with observation?

Supplemental Material

Figure S1: It seems that the two profiles have different surface temperatures, is it the case? If yes, why?

In the description of figure S2 you write: "The panel on the right shows the yield strength profile with depth when there is no melt layer...". It is the panel on the left.

"our simulations with a melt layer are not entirely self-consistent with the temperature profiles in Figure S1": that's just a detail, but I guess you can only be "self-consistent" with... yourself. I'd simply write "... are not entirely consistent with..."

"that is appropriate of molten materials...": appropriate for

"and OHNAKA rock softening model": the rock softening model

In Table S2, you specify no melt temperature for the melt layer (which I understand), but you do specify a latent heat. Is it used?

Figure S5: Do triangle indicate some topographic features that should be visible on the figure, or is it simply the position of the ring(s) taken from the plastic strain localization (as visualized in Fig. S3)? If the topography of the rings should be seen on this figure, a close up would be good.

Figure S6 (caption): Correct "tor" with "for".

Figure S7: The y axis shouldn't be negative. A negative depth is a height (and conversely, as correctly indicated in figure S8), and it's not what is shown here. Having different lines widths for the top and the bottom of the crust would help the readability.

What happens for the largest impactor and an initially 50-km-thick molten layer? The curves seem to cross.

"the final crustal signature when including a melt layer is the most similar to the GRAIL-derived crustal thickness profile (Fig. S7)": I think you're referring to figure S8.

Maxime Maurice

Reviewer #2 (Remarks to the Author):

The authors provide well thought-through arguments to my points of criticism, but I am still not fully convinced that the methodological approach is appropriate. The authors do not provide any additional data that could prove the applicability of their method. Although they now clarify in the supplementary material that the remnants of a magma ocean are approximated by a liquid layer the main text still infers that a layer of molten mantle material was modelled. In addition, my comment on the comparison between the gravity-derived crustal thickness model and the basin formation model was not addressed satisfactorily in their rebuttal.

We thank the reviewer for their comments that helped improve our manuscript. Nevertheless, we emphasize that the original manuscript noted in many places that a magma ocean was approximated as a liquid layer (the magma ocean is by definition a liquid). We also re-emphasize in our rebuttal that it would not be correct to compare our results directly to gravity data, given that the gravity signature is primarily a result of isostasy, and not crustal thickness variations. We do, however, add new simulations in support of the original conclusions. These revolve around the concerns raised over appropriateness of our numerical method.

 (1.1) The authors clarify the setup and state that their approach is thermodynamically not self-consistent in the supplementary material, but not in the main text. I think, this important information should be mentioned in the method part of the main text as I consider it as a critical simplification.

We added additional text in methods of the main text (throughout L89, 97, 101-102, 114, L145-151 and L154-161).

We expanded Section 1 in the SI to include another (hotter) temperature profile that naturally caused melting at a depth. We also made comparison simulations to demonstrate that the temperature profiles play a secondary role to the existence of a low viscosity layer under the crust. Please see our new Figure S3 in Section 1 of SI.

Nevertheless, we emphasize that we have already addressed this issue in the manuscript, where we state "Given that the composition of the melt layer is uncertain, estimating its liquidus and solidus temperatures would also be uncertain. Furthermore, the composition of the melt layer changes as the magma ocean continues to crystallize. To investigate how a melt layer would affect the basin morphology, we thus simply changed the viscosity of the material to a low, non-zero, value (100 Pa s), that is appropriate for molten materials within magma oceans while leaving the temperature of the melt unchanged."

This approach is not a "critical simplification". All that is important for our study is that the layer is molten. The composition of this layer (and hence the corresponding solidus temperature) is, in fact, irrelevant for our study.

In addition, I suggest to replace "melt layer" by "liquid" or "viscous layer" all through the text to avoid any misunderstanding.

In the abstract, we introduced this layer as: "A low viscosity layer, mimicking a melt layer,..."

I agree, that a thermodynamically consistent setup is challenging. There are many unknowns making it difficult to better constrain a more realistic setup. However, it is a major simplification and I would expect here some discussion whether this rather simple approach is justified or not.

We feel that the above text that we quote from the SI is sufficient to address this issue. Given that the composition of the molten material is irrelevant for our simulations (all that matters is the viscosity), we could in fact change the composition of the melt layer to be almost anything. If the reviewer could have provided a more concrete example describing why this is an important issue, it would have been easier for us to respond to this point.

The impact process itself generates a substantial amount of melt that would mix with the pre-impact melt. In my opinion, it cannot be ruled out that this has some effect. The authors claim that the “volume of impact melt generated remained comparable” (line 80), which does not surprise me with the given setup, but if a more thermodynamically consistent setup had been chosen, this would certainly change significantly.

We are not entirely sure which “effect” the reviewer is referring to. It is true that the impact process generates impact melt (which we account for), but this impact melt will eventually crystallize. All that is important for our simulations is the final crustal thickness profile.

Regardless, we have provided a new figure (Figure S4) that shows the region that is melted by the impact for two different simulations with greatly different initial temperature profiles. The extent of the melt (impact generated melt pool) is still insignificantly different. The reason for this is that the melt volume is more dependent on the impact conditions and impact energy put into the system than the initial temperatures, particularly at lunar impact basin size scale.

See, Fig. S4 and details in SI, section 1.

I would suggest to assume a temperature profile and solidus that generates melting in a certain depth range equivalent to the assumed liquid layer. Such a setup could be used to test whether the liquid layer approach is suitable or not. Such a test should be included in the supplementary material.

We have already addressed this comment in our responses above.

 (1.2) Generally, in my opinion, the effect of acoustic fluidization on basin formation is not well understood. Apparently, some weakening mechanism is required to reproduce the observed crater morphology, but it is difficult to justify that the mechanism of AcFl could actually work at a size-scale of basins. Well, this paper is not about AcFl and I acknowledge the text that was added to the supplementary material to justify the chosen parameters. However, the authors write in their rebuttal “Miljkovic et al. 2013 showed that basins forming in warm/hot gradients do not need to have acoustic fluidisation applied (final basin morphology forms the same in case when the acoustic fluidisation is and isn't included), whereas basins forming in a cold gradient need to have it included in the

model." I may have overlooked it, but I couldn't find anything in Miljkovic et al. (2013) showing that this assumption holds true.

Here is the relevant text in the SI of Miljkovic et al. (2013):

"Simulations employ the block-oscillation model of acoustic fluidization (46–48) to facilitate crater collapse, which is important for cooler targets. A range of acoustic fluidization parameters was tested and varied until a basin forming in a cooler target collapsed into a final basin morphology with an acceptable basin depth (5–10 km) after the simulation ended. These parameters are also similar to the ACFL parameters employed in the simulations of Chicxulub crater collapse (49). Subsequent long-term cooling and relaxation of a basin over millions of years could cause the uplift of the complete basin for another few kilometers, essentially compensating for this depth (42)."

42. H. J. Melosh, A. M. Freed, B. C. Johnson, D. M. Blair, J. C. Andrews-Hanna, G. A. Neumann, R. J. Phillips, D. E. Smith, S. C. Solomon, M. A. Wieczorek, M. T. Zuber, *The origin of lunar mascon basins. Science* 340, 1552–1555 (2013). doi:10.1126/science.1235768 Medline

46. H. J. Melosh, B. A. Ivanov, *Impact crater collapse. Annu. Rev. Earth Planet. Sci.* 27, 385–415 (1999). doi:10.1146/annurev.earth.27.1.385

47. G. S. Collins, H. J. Melosh, J. V. Morgan, M. R. Warner, *Hydrocode simulations of Chicxulub crater collapse and peak-ring formation. Icarus* 157, 24–33 (2002). doi:10.1006/icar.2002.6822

48. K. Wiinnemann, B. A. Ivanov, *Numerical modelling of the impact crater depth-diameter dependence in an acoustically fluidized target. Planet. Space Sci.* 51, 831–845 (2003). doi:10.1016/j.pss.2003.08.001

49. G. S. Collins, J. Morgan, P. Barton, G. L. Christeson, S. Gulick, J. Urrutia, M. Warner, K. Wiinnemann, *Dynamic modeling suggests terrace zone asymmetry in the Chicxulub crater is caused by target heterogeneity. Earth Planet. Sci. Lett.* 270, 221–230 (2008). doi:10.1016/j.epsl.2008.03.032"

The reference that was added to the text, Ivanov et al. (2019), does not consider AcFl at all, if I am not mistaking. The transition from a temperature regime where AcFl is required to a regime where basin formation may work without AcFl most likely depends on the temperature profile and, thus, is related to my previous comment on the thermal state and a thermodynamic self-consistency. I think, these things are coupled which is why I think the matter of AcFl cannot be excluded from the discussion.

We removed the reference to Ivanov's work and restrained the citation to our previous work only. Ivanov's work has actually demonstrated that there was no need to include the acoustic fluidisation for large craters in the first place. Our work (Miljkovic et al., 2013) showed when it was necessary to employ the acoustic fluidisation to match crustal profiles, namely topographic levels, with observations. Acoustic fluidisation in the case of lunar basin formation helps to account for the central depth of the observed basins. This actually has very little to do with the actual crustal thickness observations, which is the main focus of this study. Nevertheless, we made additional simulations to show that when a melt layer is present, the crustal profiles are almost identical (Figure S3 bottom).

 (2) The authors did not reply to my question regarding the comparison of gravity-derived models of crustal thickness in Fig. 1 and the formation model in e.g. Fig.2 or Fig. 4. In the formation model the mantle uplift has a different shape (crustal slab or melt-layer are sandwiched in between mantle material) than in the gravity-derived model where a simple dome structure is assumed. To my understanding the crustal thickness inversion only accounts for a simple two-layer case that does not account for the fact that the structure of the central mantle uplift may be more

complicated. I agree, computing the gravity signature from the formation model is not straight forward so I cannot provide a solution here, but I would expect some discussion and further explanation how the comparison is made. The stated similarity between formation model and crustal thickness is not obvious to the reader. In fact, as I mentioned already in my previous review, the gravity signature is related to the composition of the melt layer. It is nowhere stated whether the liquid layer is considered to be part of the mantle or whether it has a composition that is rather similar to the crust. I guess, if this is some end-member product of the magma ocean crystallization I presume its density may be somewhere in between the crust and mantle. I think, this should be discussed in the main text.

The reviewer asked in their previous review to compute the predicted gravity signature from our simulations and compare this with observations, instead of comparing crustal thickness profiles. We did in fact respond to this question, and we even added the following text to the SI as a result of the question:

“In our work, we compare the relief of the surface and crust-mantle interface with the known surface topography of the Moon and GRAIL-derived crustal thickness models. An alternative approach could have been instead to compare directly the gravity field predicted by iSALE with GRAIL observations. However, to do so, it would have been necessary to specify the density and porosity of the crust, which are both uncertain. Any numerical oscillations in the vertical direction at 2-3 h time in the iSALE simulations would also have affected the gravity signal. Furthermore, we note that isostatic adjustment of the basin (which is largely vertical in nature) would modify the predicted gravity field but would not affect significantly the crustal thickness profile. For these reasons, we chose to analyse the crustal thickness profiles instead of the predicted gravity.”

Given that it would be a bad idea to compare predicted gravity signatures, and that we have already responded to this question, we have not made any significant changes to the text.

Finally, a few minor points need to be revised:

In Fig 4. according to the legend the 25 km melt layer is shown, but in the caption it is stated that the melt layer was only 10 km thick.

This has been corrected to 25 km.

At line 100 it says “When no melt layer is present the crustal cap instead remained extremely thin within the peak ring...”, this only holds true for the larger impactors. For the 60 km impactor (Fig. 4) no crustal cap occurs at all. This may require some discussion.

This is correct, and rectified in the main text by adding “(or, absent⁴⁴)” in that sentence. We added the relevant reference 44 (Miljkovic et al., 2015 EPSL) to this sentence that discusses in much more detail mantle exposures and crustal caps in lunar basins.

At line 135 the authors state that the crust was modelled “...using an analytical equation of state for granite...”. This needs to be explained as the lunar crust is certainly not composed of granite. A reference may be sufficient here as it is not

uncommon to use granite, but I think, for somebody not familiar with modeling of crater formation on the Moon may be confused by this.

We added the following explanation: "These are simplifications in terms of chemical compositions of both the crust and the melt layer, however, there is a limited number of validated and widely used constitutive models for typical rocks, which is why we used the ones that are the most similar in terms of density." Miljkovic et al., 2016 JGR reported that using either basalt or granite equation of state to represent the crust did not significantly change the final basin morphology in their simulations. Unfortunately, there isn't an ANEOS for anorthosite, which is why we used substitutes. Some of previous modelling works used an adopted Tillotson equation of state for anorthosite, however, this equation is appropriate for high pressure phases of metals. Neither option is perfect. Here we decided to keep ANEOS which tracks rocks through P-T ranges and allows for phase change too.

Reviewer #4 (Remarks to the Author):

most of my concerns and comments address minor points and aim at easing the understanding to a broad audience that might not be familiar with impact modeling (myself included). While the point is clearly made and the article well written, I think that it would benefit from extending the discussion on several points. The main text is very short, so that this can be done without trespassing the journal's requirements.

We thank the reviewer for the comments that helped us clarify our manuscript for those who are not familiar with the details of iSALE simulations.

My main objection is that, in spite of criticisms raised by the other reviewers, the authors haven't made clear that the initial temperature profile plays only a secondary role. Simulations have been made with two different profiles, but no comparison is shown, and the reader must be content with the simple claim that the differences are minor, relegated to the supplemental material, and without it being supported by any quantitative information. Furthermore, it is never stated which of the two profiles is employed in the results presented. Since the authors have worked extensively on characterizing the influence of the thermal profile on the crater morphology in previous studies, I expect them to have a few more relevant things to add to the discussion. In particular because the presence or absence of a molten layer likely correlates with a specific thermal state (hot profile when a molten layer is still present and cold layer when the mantle is fully solid).

Based on the above comments, section 1 of the SI now includes updated (and new) Figures S1-S4 that address these comments.

Figure S1 now includes 3 thermal profile (Initial Moon A, B, and C), as opposed to only 2 that were previously shown ("Initial Moon C" is the new profile). This thermal profile crosses the solidus and allows for the creation of melt at depth without setting up our models with an additional melt layer (see, Figure S2 for the yield strength profiles with depth).

We also clarified where necessary in the text which thermal profiles were used when presenting our results: A was used for curved targets simulating larger basins, whereas B was used for flat target simulating smaller lunar basins. This was mainly

because of inability to apply custom temperature profiles on curved surfaces. We needed a curved surface for simulating the largest lunar basins. New Figure S3 shows crustal thickness profiles for the same impact basin forming in these three different thermal gradients (Figure S3 top compares Initial Moon A vs B, and Figure 3S middle compares Initial Moon B and C profiles). The crustal thickness profiles are seen to be insignificantly different suggesting that the selection of the thermal gradient does not play a major role when a melt layer is present.

I can imagine that the impact-induced melt pond in the crater is a transient feature whose rapid evolution is hard to accurately describe. In the present case, do you expect (or is the model implicitly solving for) interactions between it and the pre-existing molten layer? Following this line, although I understand the choice of representing the 3 different components (crust, molten layer and mantle) in Fig. 2, I wonder how relevant it is to distinguish between the molten layer (which is no longer much of a layer after the impact) from the molten part of the mantle (which is not distinguished from the mantle). How deep is the melt pond in the crater at this stage, and would it be possible to represent it on the Figure?

We believe that Fig. 2 need to show only the crustal profile, as the density difference between the crust and mantle creates the largest component of the gravity signature.

Further explanation includes a new figure in the SI that addresses this issue. Figure S4 shows the final basin morphology for the same impact conditions using two different temperature profiles (cold B and hot C). The melt pool in the basin centre is up to 200 km deep and up to 300 km in radial distance extending at the surface and under the crust. The melt pool here includes partial and complete melt, with temperatures above ~1500 K. The melt is produced primarily from uplifted mantle material. The melt layer in both cases is pushed away during excavation, and like the crust, does not contribute to the overall melt volume significantly. Some amount of melted crust or the initial melt layer could be part of the melt pool, but that would be a very small fraction. We further note that melt pools were more investigated in more detail in our previous studies (e.g., Miljkovic et al., 2015 EPSL), where we showed that the crustal contribution is minimal in lunar basin melt pools. Furthermore, we note that in our figures, the boundary between cells that are predominantly crust, melt layer or mantle are separated by thick black contours separating each material in the simulations. These boundaries were drawn by checking when material changes from a cell to cell.

Is it possible to draw conclusion on the excavation of material from the molten layer, which could have a clear compositional signature (being composed of very evolved liquids)? For instance, in the left panel of Fig. S4a, material from the molten layer seems to be excavated at the center of the crater, but this is still a very early stage, and long-term processes not addressed here might blur this signature.

The major problem is that we do not know what the composition of the melt-layer/magma-ocean was at the time of each impact. Melosh et al 2017 (Geology (2017) 45 (12): 1063–1066) have in fact argued that the highlands surrounding SPA show evidence of pyroxene rich materials that may be derived from the mantle (or magma ocean). However, our previous work combined with remote sensing data shows that basins on the nearside excavated significant quantities of olivine. The problem is that the magma ocean changes composition as it crystallizes. During

crystallization, the solid portion of the mantle can also overturn. There are so many questions related to this that we did not find it fruitful to discuss the remote sensing implications in any detail.

Nonetheless, there were no large (easily resolvable) regions where a melt layer was the predominant component on the surface. Furthermore, the amount of excavated material from the initial melt layer was minimal, with most of melt collapsing back into the crater cavity during crater formation.

Another long-term process that would be worth mentioning in the case of large basins is impact-induced magmatism in the mantle (e.g. Elkins-Tanton et al., 2004), as it could be important if the mantle is still hot, and could strongly affect the crater signature.

The reviewer notes an interesting problem in lunar science, where impact events could potentially give rise to impact induced magmatism. This topic is somewhat contentious with some arguing against such a process (e.g., Ivanov and Melosh, 2003) and others arguing for (Elkins-Tanton et al. 2004; Ghods and Arkani-Hamed 2007). Part of the controversy regards when this basaltic volcanism would occur (immediately after basin formation, or hundreds of millions of years later). We don't feel that our simulations bring any particular insight to this problem. In fact, given that our simulations start with a molten layer beneath the mantle and crust, we simply note to the reviewer that this would make it more difficult for basaltic magmas to traverse this melt layer and erupt on the surface.

Finally, being a bit more exhaustive in terms of which of the ancient craters might be concerned by this enhanced relaxation, and whether one can see a transition (and a potential link between the crater stratigraphic ages and the magma ocean lifetime), as was done for instance by Conrad et al., 2018, would be an interesting addition.

We note that we do cite the paper by Conrad et al. (2018) in the main text, who showed that the oldest basins are more relaxed than the younger ones. We think that the reviewer is asking whether we can date when our "craters that are degraded beyond recognition" formed. Unfortunately, we don't think that this is possible, even though it would be very useful if we could. We note that our degraded morphologies form when a melt-layer or magma ocean is present, and that our models do not really place any constraints on when this might have been. Also, the argument is difficult to address, because we argue that basins that formed at this time would be so degraded that we probably wouldn't even recognize them. We have not modified the text because of this, but we appreciate the question which led us to reflect on how we might try to address this important issue. Unfortunately, the timing of when a melt-layer/magma-ocean was present will need to come from independent geochronological studies.

Main text

L. 50: "age criteria P-13 and P-14": I think that if this age criterion is to be mentioned, it deserves a minimum of precision (merely referring to the literature distracts the unfamiliar reader).

We agree that the “P-13” nomenclature will not be clear to many readers. We have simply decided to remove these from the text in order to improve the readability.

L. 53: “between one and two radii from the basin center”: Can you explicit how the crater diameter is defined? (is it the topographic ring diameter as indicated in the caption of Fig. 1?)

Clarifications added in the main text: L52-L55 plus, we now state that we are using previously mapped main topographic rim diameters. Here is the revised text for this section that should make this clearer:

“For example, Fig. 1 (top) shows three of the oldest pre-Nectarian basins^{1,35} that exhibit less prominent crustal thinning compared to younger basins of likely similar size, such as the Nectaris and Orientale basins (bottom). Though the sizes of these basins are similar, based on the diameter of their previously mapped main topographic rings, the older pre-Nectarian basins have a relatively thicker crust in the centre of the basin and also lack the distinct crustal thickening between 1 and 2 main rim diameters as is observed within younger basins^{17,36}”

We have furthermore changed the caption to more clearly explain the origin of the plotted ring diameters:

“Arrows denote previously mapped main rims (observed or suspected) whereas the squares in the bottom panel correspond to the basin’s peak (inner) ring³⁴”

We also removed ring locations from Fig S7.

We removed D_{thin} info from Table S1 as we only refer to ring locations. Therefore, Table S1 has been slightly edited.

We updated Figure 1 to include a suspected ring of Fecunditatis that wasn’t there before.

L. 84: write “depend” (without “s”)

changed.

L. 98-99: “Furthermore, when a melt layer is present, the thickness of the crustal cap in the centre of the basin is larger than in the case where there is no melt: It seems that for the Orientale-like impact, there is no crustal cap at all at the centre.

This indeed appears to be the case. There are four possible ways to reconcile this with the observation that the mantle is not exposed in the Orientale basin. First, Orientale could have formed with thermal gradients that were not as cold as the one used here, and this would help to form such a crustal cap. Second, the melt pool in the centre of the Orientale basin could have differentiated forming a secondary crust. Third, it is possible that Orientale does not have a crustal cap: Crustal thickness models suggest that the crust is in fact very thin in this basin and the crustal thickness models do become less accurate the further you are from the Apollo seismic stations. Finally, it is possible that the mantle was exposed at the surface, but that the subsequent volcanism obscured these deposits. Given that this is not the main focus of this work, we have not made any changes to the text.

Fig 2. For the SPA-like case without liquid layer, there seems to be a significant layer of mantle ejecta covering the crust as far as >700 km (even more with the curvature) from the center, while the crust remains exposed in the case with a liquid layer. How does it compare with observation?

The reviewer notes that our simulations for SPA predicts that much of the ejecta that lands exterior to the basin is derived from the mantle, and wonders whether we would see a geochemical signature of this. This is a complicated question that goes far beyond the scope of this manuscript, but we can note a few things. First, Melosh et al 2017 (*Geology* (2017) 45 (12): 1063–1066) in fact argue that the highlands surrounding SPA show evidence of pyroxene rich materials that may be derived from the mantle. Second, we note that if SPA formed when a magma ocean was present, that the composition of the ejected materials would be highly sensitive to how much of the magma ocean had crystallized at that point. Lastly, we note that mantle overturn could also complicate the question as to what the composition is of the upper mantle: In particular, it has been argued that the mantle has been excavated from several basins and that the composition of this material is close to that of a dunite. Though we have no answer for the reviewer, it should be clear that we don't know what compositional signature we should expect.

Supplemental Material

Figure S1: It seems that the two profiles have different surface temperatures, is it the case? If yes, why?

The thermal profiles B and C use a 250 K surface temperature whereas the thermal profile A uses 80 K. The higher value is representative of what one might expect near the equator, whereas the lower value would be more appropriate closer to the poles. However, even with such low starting temperature, the near surface temperature rises quickly to 250 K within top 4 km given a steep gradient of 50 K/km that was applied. We note that our simulations are very similar for all three profiles, so our results are not sensitive to the exact chosen surface temperature.

In the description of figure S2 you write: "The panel on the right shows the yield strength profile with depth when there is no melt layer...". It is the panel on the left.

Corrected and sentences swapped for clarity.

"our simulations with a melt layer are not entirely self-consistent with the temperature profiles in Figure S1": that's just a detail, but I guess you can only be "self-consistent" with... yourself. I'd simply write "... are not entirely consistent with..."

corrected.

"that is appropriate of molten materials...": appropriate for

corrected.

"and OHNAKA rock softening model": the rock softening model

corrected.

In Table S2, you specify no melt temperature for the melt layer (which I understand), but you do specify a latent heat. Is it used?

No latent heat calculations are included in iSALE simulations to date (This is not easy to fix and it's been a priority of the developers for some time now). This is part of the reason for using initially lower temperature gradients in order to account for a potential overestimation of the quantity of melt that is generated. Other studies worked around the latent heat issue, such as Potter et al.'s various works including his PhD thesis.

Figure S5: Do triangle indicate some topographic features that should be visible on the figure, or is it simply the position of the ring(s) taken from the plastic strain localization (as visualized in Fig. S3)? If the topography of the rings should be seen on this figure, a close up would be good.

Our Figure S5 deals with ring formation, whereas Figure S7 shows the dependence of basin morphology on the melt layer. Not all runs use the same resolution; The full extent of topographic multi-rings (shown in Figure S5) requires very high-resolution simulations which was not the case in Fig. S7 (former Fig. S5). Therefore, we removed the triangles from this figure to prevent confusion.

Figure S6 (caption): Correct "tor" with "for".

Corrected.

Figure S7: The y axis shouldn't be negative. A negative depth is a height (and conversely, as correctly indicated in figure S8), and it's not what is shown here. Having different lines widths for the top and the bottom of the crust would help the readability.

Thanks for helping us clean up the plots (Figs S8-S9): depth below the surface is indeed a positive number.

(continued...) What happens for the largest impactor and an initially 50-km-thick molten layer? The curves seem to cross.

The crustal profiles are composed of separate curves: one that tracks the crust-mantle interface and the other that tracks crust to free surface boundary. In some cases, when the crust is absent, the two curves converge into a single curve, or fail to show surface level if mantle is exposed to the surface. This has been remedied, and we have ensured that the surface now extends all the way to the center of the basin (Fig. S9).

"the final crustal signature when including a melt layer is the most similar to the GRAIL-derived crustal thickness profile (Fig. S7)": I think you're referring to figure S8.

Corrected.

Additional improvements:

Page 6, grammar improved

Fig. 3. Removed arrows from plot and added distances in the caption.

Fig. S10. Extended the caption.

Vertical axis is labelled as z . The surface level is zero and goes positive above the surface. This is why often depth was mistakenly in sub-zero values. To avoid further confusion, z has remained as vertical axis pointing above the surface, and if depth was used to label the vertical axis, the values were positive.

REVIEWERS' COMMENTS

Reviewer #4 (Remarks to the Author):

The updated version of the manuscript addresses satisfactorily my major criticisms. The motivation for leaving some of the points I raised (which were more open questions than actual objections) is convincingly presented in their rebuttal. I recomand that the paper be accepted for publication.